# Corporate Criminal Liability: An Overview of the Croatian Model after 20 Years of Practice †

**Igor Vuletic**

Faculty of Law, Josip Juraj Strossmayer University of Osijek, 31000 Osijek, Croatia; ivuletic@pravos.hr
† The research for this paper was conducted within 'Artificial Intelligence and Criminal Law (IP-PRAVOS-18)', a project funded by the Faculty of Law Osijek.

**Abstract:** The Croatian legislators introduced the concept of criminal liability for legal entities already in 2003 with the adoption of the Law on Criminal Liability of Legal Entities. Influenced by the writing of esteemed domestic scholars, and inspired by French law, the legislators opted for a system linking the liability of corporations to the liability of the responsible person. There were very few cases in practice during the first years of its application, and the situation changed after the first prominent indictment of this type against the ruling political party for economic crimes. Since then, the legislation has been amended several times and a significant body of jurisprudence has developed. In the first part of this paper, I will describe the chronology of the development and formation of the Croatian legislative model of corporate criminal liability. The second part will analyze 31 available final court judgments, which will be the basis for the conclusion about the issues in the practical application of the legislative model and, more generally, the phenomenon of criminal offenses committed by legal entities in Croatia. Based on this analysis, I will indicate the potential deficiencies of such a concept. In the context of future development, special attention will be given to the problem of economic crimes committed by AI corporate systems.

**Keywords:** sanction; guilt; artificial intelligence; corporation; culpability

## 1. Introduction

The first indications of a need for criminal liability of legal entities appeared in the early eighteenth century, following the failure of the South Sea Company, which was one of the largest companies at the time, and the subsequent financial crisis which resulted in enormous economic losses (Engelhart 2014). These events contributed to public awareness of the fact that the actions of corporations can cause significant and long-term consequences and that it is not sufficient to convict a natural person in order to provide adequate criminal law protection. The concept of corporate criminal liability is generally accepted, despite the fact that a small number of authors argue for exclusive civil liability, emphasizing the negative economic effects and the stigmatizing effect of criminal proceedings (Byam 1982; Arlen 1994; Khanna 1996). It has been developed in the Anglo-American legal systems for over a century, while it has been established more intensively in continental Europe in the past few decades (Beale and Sawfat 2004). This is primarily due to the fact that most national criminal laws of continental Europe are based on a rigid understanding of the concept of liability, which is much different than the more flexible concept of mens rea from the Anglo-American legal tradition (Mueller 1957).

Criminal liability for legal entities was introduced into Croatian law in 2003, with the adoption of the Law on the liability of legal persons for criminal offenses. This law was met with public approval, which was probably due to the fact that the Croatian economy in the 1990s was pillaged and destroyed by transition and privatization crime. In the post-war period in the mid-1990s, a significant part of the industry sector was privatized and destroyed, which led to a massive loss of jobs and lifestyles (Gregurek

2001). The perpetrators of these offenses were usually hidden behind some legal entities, which naturally caused public dissatisfaction. Therefore, the adoption of this Law was significant not only from the legal, but also the social and political perspectives. This was amplified by the fact that the Law was adopted after the conclusion of the Agreement of Stabilization and Accession to the EU, which initiated a number of expansive reforms in the country (Derenčinović and Novosel 2012). The new Croatian legislation was inspired by the French Criminal Code, the German Law on Misdemeanors and the Slovenian Law on the liability of legal entities for criminal offenses (Đurđević 2003). With this big reform, the Croatian law adapted to the new socio-political-economic trend of European, and especially post-transition, countries (Lederman 2000).

In the first years since its enactment, the Law did not see a significant rate of application. The proceedings against legal entities were very rare, partially due to the fact that the entire concept of criminal liability for legal entities was a novelty that seemed controversial from the perspective of the conventional criminal rulebook. The traditional notion of societas delinquere non potest was still deeply rooted in the minds of practitioners (Đurđević 2003). It can be argued that the legislator expected such a situation, given the atypically long vacatio legis of six months, instead of the usual eight days, leaving sufficient time for the adjustments to the new framework. Since then, the Law has been amended five times. However, the jurisprudence started developing more significantly after 2011 and the initiation of the proceedings against the leading political party and the former prime minister Dr. Ivo Sanader for corruption and economic crime (Vuletić 2014).

Today, a full twenty years after the enactment of the initial version of the Law, criminal proceedings against legal entities are no longer an exception. Based on the experience from its application, several legislative amendments and the proper standards of interpretation were formed. Therefore, Croatian law is suitable for scholarly analysis because it can serve as a useful example for the legal systems that have not yet developed the concept of criminal liability for legal entities.

This paper will outline the Croatian legislative model and the experiences in its practical application. The first section will describe the legal concept of criminal liability of legal entities, while the second section will illustrate the situation in practice. This will be based on the statistical indicators on the number of proceedings, the types of decisions and the phenomenology of criminal offenses for legal entities. Furthermore, I will present the results of my own research on the main challenges in criminal proceedings against legal entities, based on a sample of thirty-one final decisions. Based on the described analysis, I will offer a conclusion on the success of the Croatian model of criminal liability of legal entities.

## 2. Croatian Legislative Model

There are several categorizations of the models of criminal liability for legal entities. This topic has been covered in high-quality scholarly work, which comprehensively presents the situation in comparative law (Cavanagh 2011; Clough 2007; Engelhart 2014). As this paper is focused primarily on the experiences of the Croatian legal system, an exhaustive discussion of comparative models would exceed the scope of the analysis. Therefore, this theoretical introduction will present only the basic outlines, primarily from the perspective of the Croatian literature as it has impacted the development of the Croatian legislative solution.

In the Croatian theory of criminal law, there is a common division of liability into the objective model, the establishment of the liability of the legal entity based on the liability of the responsible persons and the autonomous liability model (Novoselec 2016; Đurđević 2003). According to the objective model (vicarious liability), which is characteristic to the Anglo-American legal systems (Lederman 2000; Novoselec 2016), there is no need to establish the liability of the legal entity, just the fact of harm caused to protected goods, which exceeds the boundary of acceptable risk. As such, the legal entity will be liable for the acts of its employees because they act in its name and for its account (Weissmann 2007).

The objective model is characteristic of the U.S. legal system with its principle of *respondeat superior*. In contrast, the English common law approach employs the identification principle whereby the liability of the legal entity is based on the liability of the responsible persons at the "directing mind and will" level. (See, e.g., the Law Commission of England & Wales, Corporate Criminality, Discussion Paper 2021 for a good overview). According to the other model, typical for European continental legal systems (Lederman 2000), the liability of a legal entity is a prerequisite for its criminal liability, and it is derived from the liability of the responsible natural person in charge of the operations of the legal entity. According to the third concept, which is of a purely theoretical nature and not widely accepted in comparative legislation, a legal entity can bear its own, autonomous liability based on the fact that it has breached some moral norm of existential social importance. This would be a sort of liability in the broader sense, due to a poor organization of work. Some contemporary theories follow this line of reasoning, claiming that the legal entity should be considered a "legal container" because of its high level of autonomy in establishing the structure and hierarchy of the organization. It is capable of taking independent rights and obligations on the market, and therefore also bearing independent liability (Bayern 2016).

The Croatian legislators, as a representative of the typical European continental system, accepted the model of liability for legal entities based on the liability of their management bodies, i.e., the responsible natural persons. Thus, Article 3 paragraph 2 of the Law provides that the basis of criminal liability of legal entities is the criminal act of the responsible persons. The relevant act needs to be a crime and not a misdemeanor or a breach of ethical rules, which are not criminal. If there is no such criminal act, there is no basis for criminal prosecution and punishment of the legal entity. In the practical sense, this means that there will be no criminal liability of a legal entity if the responsible person acted under some exception from criminal liability (e.g., out of necessity) or if the responsible person is not liable for any reason (Article 5 paragraph 1 of the Law). According to the Croatian criminal code, liability is excluded in the cases of unaccountability, and in some situations of errors in fact or law. However, there will be no need to prove the liability of the legal entity separately, but it will be automatically presumed as established (praesumptio iuris et de iure) once the liability of the responsible person is established (Supreme Court of Croatia, I Kž 300/2011; Novoselec 2016). Criminal liability covers all entities which, according to the provisions of the applicable Croatian laws, hold the status of legal entities. This includes foreign legal entities, but it does not include the Republic of Croatia as a State (Article 6 of the Law).

The responsible person is, primarily, a natural person who is in charge of managing the operations of the legal entity. This category includes directors, management members, members of oversight boards, etc. However, any other natural person who is entrusted with conducting work within a specific legal entity can also be considered a responsible person (Article 4 of the Law). Special legislation provides the specific persons in charge of managing the operations for each type of legal entities. The concept of responsible persons is not limited to management structures. Namely, according to Croatian judicial practice in this respect, it is not crucial whether there is a legal basis for the assignment of such duties (such as a decision of the competent body or power of attorney), but it is relevant whether the person actually performs the duties. In line with this position, courts have treated as responsible persons a journalist working for a journalistic corporation based on an employment agreement (Supreme Court of Croatia, I Kž-Us 15/2021), a truck driver in a transport company (County Court in Šibenik, Kž 192/2022), a university professor at a faculty (Supreme Court of Croatia, I Kž-Us 51/2020), an excavator operator in a construction company (County Court in Bjelovar, Kž 335/2021), etc. In other words, the concept of a responsible person is interpreted very broadly, which enables its application to any person who is effectively in a position wherein they perform a certain duty within the competence of the specific legal entity under trial, even in situations when this person is not an employee of the legal entity but is engaged only for particular work (Supreme Court of Croatia, III Kr 65/2022).

In order for criminal liability to exist, one of the following two alternative requirements must be met. The responsible natural person must violate one of the duties within the competence of the legal entity (for example, the duty to protect the workers and pedestrians on a construction site, the duty to ensure measures against environmental pollution in a factory, etc.) or the legal entity must obtain illegal benefits for itself or third persons through a concrete criminal offense (Article 3 paragraph 1 of the Law). With regards to the first requirement, judicial practice has taken the position that it is necessary to determine the concrete duty of the legal entity that has been violated, in what way and whether there is a causal link between the violation of such a duty and the consequence. In other words, the violation of the duty of the legal entity will not be automatically presumed if the liability of the responsible natural person is determined. If the supervisor of a construction site failed to provide safety equipment to workers, leading to the injury and death of a worker, this does not automatically entail the liability of the construction corporation if it is determined that it acquired such equipment and gave it to the supervisor to distribute to the workers (County Court in Zagreb, 1. Kž 760/2017). With regards to the second requirement, there is a distinction between situations in which the responsible person obtains financial benefits for themselves and the legal entity, versus situations in which the financial benefits are only in favor of the responsible person. Namely, the liability of the legal entity will only be possible in the former situation, while in the latter situation, the criminal liability will be borne only by the responsible natural person, while the legal entity can appear in the proceedings as a damaged party, or not at all (Novoselec 2016). This position is accepted in Croatian judicial practice as well, thus a defendant corporation was released from liability in a situation where its director borrowed money from the corporation, which she never returned, and erased the loan from the corporate records (Supreme Court of Croatia, I Kž 298/2014).

In the procedural sense, the Law provides that the legal entity and the responsible person will be tried in a single procedure and that a single judgment will be issued for both (Article 23 paragraph 1 of the Law). An exception to this rule is possible only if there are certain legal or factual barriers to the trial (for example, death, lack of legal capacity, immunity, a situation in which it is not possible to determine the responsible person, etc.). In such instances, the proceedings will only be held for the legal entity (Article 23 paragraph 2 of the Law). However, even in such cases, the court will have to declare in the dispositive of the judgment that the criminal offense was committed by the responsible person, but that the trial could not take place against them for specific reasons (Novoselec 2016).

With regard to the prescribed criminal law sanctions, the Law provides punishments (based on liability) and security measures (based on the risk level of the offender at the time of sentencing). There are two types of punishments, which are the termination of the legal entity (as the strictest punishment) and monetary fines. The termination of the legal entity is an exception in practice and monetary fines are the regular course of action (Derenčinović and Novosel 2012). Their amount depends on the duration of the imprisonment prescribed for natural persons for the relevant criminal offense. The lowest possible fine is EUR 660.00, and the highest is EUR 1,990,840.00 (Article 10 of the Law). It should be noted here that the calculation of fines in Euros was introduced with the most recent amendment from 2022, since the Republic of Croatia has adopted the Euro as the official currency from 1 January 2023 onward. Prior to then, fines were prescribed and issued in the previous official currency, the Croatian Kuna. Therefore, the overview of the judicial practice in the second section of this paper will express monetary fines in the previously applicable currency.

The Law provides the possibility of conditional sentencing (Article 13 of the Law), as well as the so-called effective remorse, which enables the release of a person who reports the criminal offense before they are discovered, or before they realize they were discovered (Article 12.a of the Law). This does not affect the conviction but only opens up the possibility for the court to either mitigate or fully release the defendant from sentencing. The latter option is not infrequent in Croatian criminal law, and it has the criminal-political purpose of stimulating the perpetrators to contribute to the discovery of the offense and

the reduction of its harmful consequences (Turković et al. 2013). Among security measures, legal entities can be sentenced to the prohibition of performing certain operations or work (Article 16 of the Law), prohibition from obtaining permits, concessions and subventions (Article 17 of the Law) and the prohibition of conducting business with the users of the state or local budgets (Article 18 of the Law). All three measures can be issued in the duration from one to three years from the moment of the judgment's entry into force.

### 3. Overview of the Situation in Judicial Practice

The following sections will present the current situation in Croatian judicial practice exactly twenty years after the enactment, and nineteen years since the entry into force of the Law. This period was sufficient for the development of a significant and representative body of judicial practice. This allowed the identification of the most important problematic practical issues, and the courts developed jurisprudence on how such issues should be resolved. The first part of this section will provide the statistical data from the records of the State agency for statistics of the Republic of Croatia and the State Attorney's Office of the Republic of Croatia. The analysis of this data will allow us to draw conclusions on the structure of the criminality of legal entities in Croatia. The second part will present the results of my research based on a sample of thirty-one final judicial decisions in criminal proceedings against legal entities. The third part will offer implications for potential future research in the field.

### 3.1. Statistical Overview

The State Attorney's Office of the Republic of Croatia regularly publishes annual work reports, which include the criminality of legal entities. According to the most recent Report from 2022, the rate of criminal reports against legal entities out of the total number of reports has not significantly changed between 2017 and 2021.[1] This can be observed in Table 1:

**Table 1.** Criminal reporting of legal entities.

| Year | Number of Reports | Percentage of the Overall Criminality |
|------|-------------------|---------------------------------------|
| 2017 | 1263 | 3.27% |
| 2018 | 1188 | 3.37% |
| 2019 | 1165 | 3.24% |
| 2020 | 1345 | 3.36% |
| 2021 | 1857 | 4.51% |

It is visible that the number of criminal reports in the observed five-year period is almost even and that the rate of criminality of legal entities in the overall criminality column is relatively low, ranging between 3.24 and 4.51%. However, when comparing 2021 to previous years, it is actually up by 50% (albeit based on the value of the fines imposed, the gravity of the offending behavior is relatively low). The Report shows that the largest number of criminal reports relates to economic crimes. Furthermore, the analysis of the grounds for rejecting criminal reports shows that 62% of rejected claims were due to the absence of liability of the responsible person, or because of the termination of the legal entity.

The State Agency for Statistics maintains and publishes statistical data on the trends in the number of reported, indicted and convicted natural persons and legal entities. The Table 2 below shows the trends of the number of reported, indicted and convicted legal entities in the five-year period between 2016 and 2020:

---

[1]  Available at: https://dorh.hr/hr/izvjesca-o-radu (accessed on 19 January 2023).

Table 2. Reported, indicted and convicted legal entities in the period between 2016 and 2020.

| | 2016 | 2017 | 2018 | 2019 | 2020 |
|---|---|---|---|---|---|
| **Reported** | 1028 | 883 | 758 | 768 | 707 |
| **Indicted** | 169 | 121 | 99 | 77 | 86 |
| **Convicted** | 59 | 31 | 31 | 31 | 49 |

Source: Croatian Bureau of Statistics. Available at: https://podaci.dzs.hr/2021/hr/10009 (accessed on 19 January 2023).

These numbers are somewhat different from the presented data of the State Attorney's Office of the Republic of Croatia. It can be noted that the number of criminal reports and indictments in the last observed year has significantly decreased in comparison to the first year. On the other hand, the number of convictions, while lower than in 2016, increased in 2020 as compared to the period between 2017 and 2019. The Table 3 shows the structure of criminality for legal entities based on the issued judicial decisions (held liable or not liable) in 2020:

Table 3. Structure of the criminality of legal entities in 2020.

| | Criminal Offenses in Employment Relations | Criminal Offenses against the Environment | Criminal Offenses against the General Security | Criminal Offenses against Property | Economic Crimes | Intellectual Property Crimes | Criminal Offenses against Official Duties | Other Criminal Offenses | Total |
|---|---|---|---|---|---|---|---|---|---|
| **Liable** | 6 | 3 | 6 | - | 33 | - | - | 1 | **49** |
| **Not liable** | 5 | 2 | 5 | 5 | 17 | 1 | - | 2 | **37** |
| **Total** | 11 | 5 | 11 | 5 | 50 | 1 | - | 3 | **86** |

Source: Croatian Bureau of Statistics. Available at: https://podaci.dzs.hr/2021/hr/10009 (accessed on 19 January 2023).

The data show that there were 86 judicial decisions in 2020 that concluded proceedings against legal entities. Out of this total number, a slightly higher rate (57%) were convictions, while the remaining decisions were not (the charges were dismissed, the charge was rejected because of the lapse of the statute of limitations or the trial was discontinued due to the death of the responsible person, the termination of the legal entity or other reasons provided by law). In the structure of the criminal offenses for which legal entities were on trial, economic crimes dominate (58%), followed by criminal offenses against general safety and employment relations (13% each). Other criminal offenses are much less frequent. The prevalence of economic crimes was confirmed in the comparative research on the status of criminality of legal entities from 2013 (Derenčinović and Novosel 2012).

When it comes to sentencing, the data of the State Agency for Statistics show that only monetary fines were ordered and that there were no terminations of legal entities. Most of the monetary fines ranged from HRK 10,001 to HRK 20,000 (21 legal entities), followed by those between HRK 20,001 and HRK 50,000 (20 legal entities). In five cases, the court ordered the confiscation of financial benefits obtained through the criminal offense.

*3.2. Overview of My Research*

I have conducted my own research on the Croatian judicial practice on a sample of thirty-one final judicial decisions issued in criminal proceedings against legal entities during the period between 2013 and 2022. I have collected the decisions through a search of the publicly available portal SupraNova.[2] The "SupraNova" portal is the information

---

2 Available at: https://sudskapraksa.csp.vsrh.hr/home (accessed on 9 January 2023).

system used in all regular and special courts and the Supreme Court of the Republic of Croatia. The system includes all the final decisions of these courts.

The following Table 4 shows the structure of criminal offenses for which legal entities were tried in the judgments covered by this research.

**Table 4.** The structure of the criminality of legal entities.

| Type of Criminal Offense | Number of Judgments | % |
|---|---|---|
| Against the economy | 21 | 68 |
| Forgery of documents | 1 | 3 |
| Against general safety | 5 | 16 |
| Other criminal offenses | 4 | 13 |
| **Total** | **31** | **100** |

There is a visible prevalence of economic crimes (68%), followed by criminal offenses against the general safety (16%). This ratio largely corresponds to the data of the State Agency for Statistics from the previous section. It can be concluded that all the criminal offenses from the table are connected to the performance of certain work that is part of the operations of legal entities.

The following table lays out the structure of judicial decisions in the analyzed cases based on whether the legal entity was held liable or not. For a better understanding, it should first be noted that there are three types of possible judgments. A conviction is issued by the court if it determines that the defendant committed the criminal offense they are charged with (Art. 455 Par. 1 of the Criminal Procedure Code). An exoneration is issued if the relevant act is not a criminal offense, if there are circumstances which preclude liability and if it is not proven that the defendant committed the offense in question (Art. 453 of the Criminal Procedure Code). Finally, a judgment rejecting the charges is issued by the court under certain circumstances exhaustively listed by law that preclude criminal prosecution, such as the statute of limitations, the existence of a prior judgment in the same matter, the lack of a motion by the authorized prosecutor, etc. (Art. 452 of the Criminal Procedure Code). Furthermore, in certain situations (such as death or the termination of the legal entity), the proceedings will be discontinued by a decision of the court. According to the conducted research, the structure of the judgments by type is as follows in the Table 5.

**Table 5.** Overview by type of decision.

| Type of Decision | Number of Decisions | % |
|---|---|---|
| Conviction | 15 | 48 |
| Exoneration | 7 | 23 |
| Rejection | 6 | 19 |
| Discontinuance | 3 | 10 |
| **Total** | **31** | **100** |

It appears that fewer than half of the judgments (48%) are convictions, which is different from the data of the State Agency for Statistics for 2020, which found that more than half of the judgments were convictions. It is interesting that the termination of the legal entity was ordered in one case, while in all other cases the court ordered monetary fines ranging between HRK 7000.00 and HRK 1,500,000.00. It is notable that the amount of the monetary fine depended on the scope of the consequences of the criminal offense and the amount of the illegal financial benefits obtained. Thus, the highest fines were ordered for offenses that resulted in death (offenses against general safety, mostly linked to

accidents of workers on construction sites that resulted in death) and offenses that resulted in significant financial gains (over the amount of HRK 60,000.00).

Exonerating judgments were reached almost exclusively based on a lack of evidence indicating that the legal entity had committed the criminal offense that it was charged with in the indictment. This conclusions in all cases were automatically linked to the finding of the court that there was insufficient evidence of the liability of the responsible person, which shows that the key factor for the determination of liability of a legal entity is the liability of the responsible person. Only in one case was the exoneration based on the fact that the indicted act was not a criminal offense.

Judgments rejecting the charges were issued based on the lapse of the statute of limitations and due to previous final judgments in misdemeanor proceedings against the legal entity based on the same facts (ne bis in idem). Proceedings were discontinued in situations wherein bankruptcy proceedings for the legal entity were concluded and the legal entity was erased from the judicial registry, thus ceasing to exist. In one case, the proceedings were discontinued due to the death of the responsible person during the trial.

In some cases, it was clear that the responsible person was liable, but the concrete responsible natural person could not be identified. In practice, these cases were resolved by dismissing the charges against the responsible person but nevertheless convicting the legal entity. The reasoning behind this conclusion is that it is sufficient to determine a breach committed by the responsible person to find the liability of the legal entity. If the identity of the responsible person cannot be determined, this does not affect the liability of the legal entity. This would be the case, for example, if the manager of a construction site who was responsible for the lack of safety measures was on long-term sick leave at the time a worker was injured by a fall from an unsecured scaffolding and his replacement could not be determined since there was no written decision on this matter (County Court in Varaždin, No. 5. Kž-401/2019).

The presented research shows that the criminal liability of legal entities in practice is primarily aligned with the liability of the responsible person in charge of the operations of the legal entity. It must be proven that there was a duty of the legal entity relating to a certain act or omission, that the natural person in charge of this act or omission breached their duty and that this breach caused a consequence from the body of the criminal offense. It can thus be concluded that the liability of the natural person is the key prerequisite for the conviction of the legal entity in the legislation and in practice. The only partial exception is a finding wherein a certain task falls within the competence of a natural person but their identity cannot be determined. In practice, the death of the responsible person has also led to the discontinuance of the proceedings against the legal entity, which also indicates the inseparability of the liability of the natural person and the legal entity. This fact results in the considerably high number of non-convictions (exonerations, rejections and discontinuations). This raises the question of whether such a model is sustainable in the long-term, taking into account the development of new technologies and the emergence of autonomous artificial intelligence in the sphere of the operation of legal entities. This issue is particularly relevant in light of the phenomenology of criminal offenses by legal entities reflected in this research, which predominantly consists of economic crimes. This potential issue will be addressed in the following sections.

## 4. Implications for Future Research: Corporate Criminal Liability in Terms of AI Technology?

During the past decade, the development of modern technologies has taken up a new dimension thanks to the emergence of artificial intelligence into different spheres of life and society. From the arms sector to industrial production, road transport and medicine to financial business, different artificial intelligence systems have been developed with the purpose of enhancing the efficiency of operations (Tek-Tai 2020). Meanwhile, the level of independence of artificial intelligence is increasing. Considering the fact that such systems could make mistakes in their operations, and that such mistakes can lead to smaller or

larger consequences for humans or property, this raises the question of criminal liability. This issue has been increasingly recognized in the criminal law literature, primarily in the context of autonomous driving (Gless et al. 2016; Prakken 2017; Markwalder and Simmler 2017) However, some works in the literature mention the issue of liability for financial crimes committed by artificial intelligence, warning that the concept of economic crimes based on liability and intent of natural persons will soon be inadequate to cover all potential situations of economic criminality (Yeoh 2019).

In the past several years, the emergence of artificial intelligence in the financial sector has been intensifying because automated software enables increased business profit. At the same time, autonomous algorithms assume even more risk and do not make assessments of acceptable risks on the same basis as humans (Borch 2022). Therefore, the use of artificial intelligence in the financial sector unquestionably has numerous advantages as well as risks. One of these risks relates to criminal liability for fraud and similar criminal offenses. Criminology literature warns of the possibility of abuse that can occur due to the involvement of AI in the financial sector, especially through methods such as market manipulation, price fixing and collusion (King et al. 2020). These methods imply the participation of an AI system designed to perform search tasks instead of people (Autonomous Trading Agent). Problems arise when such systems, with the ability to learn from their surroundings and received data, start emitting information with the purpose of intentionally misleading the contracting party down the wrong path. Some research has shown that such AI could master techniques of sending fictitious orders (which will never be performed) and concluding fictitious transactions with the aim of defrauding good-faith third persons and gaining profit. This could occur due to the fact that AI is programmed to, among other things, find the most profitable business models. Therefore, an AI program could recognize the conclusion of fictitious transactions as the most profitable option and then operate accordingly. Furthermore, there is the possibility of various types of illegal manipulations on the stock market through the dissemination of false information on the value of shares by algorithmic trading agents.

In the recent past, there have been drastic examples of autonomous trading algorithms causing severe financial impacts, such as the "Flash Card" incident from 2010. The interaction of several such algorithms caused significant financial losses for several subjects (Borch 2022). Furthermore, on the morning of 1 August 2012, a new (flawed) trading software sent Knight Capital Group's 7 million dollars' worth of stocks on a buying spree. Under stock exchange rules, Knight was required to pay for these shares three days later but was unable since this trade had been unintentional and lacked any financing background. Knight consequently struggled to cancel the trades, but this effort was rejected as it was considered unfair to trading partners (for the majority of stocks) by the Securities and Exchange Commission (SEC). This triggered a financial disaster for Knight, causing a 460 million dollar financial loss, which resulted in a merger with Getco LLC (Kirilenko et al. 2017).

The operation of such autonomous systems is regularly connected to certain financial corporations that use them for their functions. In most systems around the world, including Croatian criminal law, economic crimes are conceived of as intentional acts, which means that they cannot be committed in negligence. Furthermore, in Croatian law, these are mostly offenses for which the judicial practice exclusively accepts direct intent, in the sense of knowledge and intent to commit the criminal offense. It was previously stated that this form of liability must be proven for the responsible natural person, which serves as the basis for the liability of the legal entity. However, if the financial business is predominantly conducted through autonomous algorithms, it will become practically impossible to determine the liability of responsible natural persons because they will no longer be involved in the process. Therefore, it is my position that the existing concept linking the liability of the legal entity to the liability of natural persons will no longer be adequate, but that there will be a need to develop new models of autonomous liability for legal entities. In my view, this will be the most important topic of future research in the field of criminal liability of legal entities.

## 5. Conclusions

This paper presents the Croatian model of criminal liability of legal entities. In 2003, Croatia enacted a special Law on the liability of legal entities for criminal offenses, through which it implemented the concept of liability based on the determination of the liability of the responsible person. This Law was inspired by the examples from French, Slovenian and partially German law, and it has been amended several times since its enactment. There was no significant application of the law during its first years, but a consistent judicial practice has since developed.

The analysis presented In this paper showed that the criminality of legal entities makes up between 3 and 5% of the total criminality in the Republic of Croatia. Criminal proceedings against legal entities are mostly conducted for economic crimes, criminal offenses against general safety and employment relations. Other criminal offenses, such as those against the environment, are less prevalent in the official records.

Furthermore, the analysis of judicial practice shows that the main issues in the determination of criminal liability of legal entities relates to the absence of (or inability to prove) the criminal liability of the physical persons who are responsible for the legal entity. In addition, convictions do not occur if the legal entity was already criminally convicted for the same facts (ne bis in idem) and if it was already erased from the judicial registry (mostly due to the conclusion of bankruptcy proceedings), which makes it de jure non-existent. Thus, the determination of liability in practice always depends on the ability to prove the liability of the responsible persons and the violation of a duty of the legal entity. The only exception is the inability to identify the concrete responsible person.

When it comes to sanctions, Croatian law provides a monetary fine and the termination of the legal entity as options. The analysis of the situation in the judicial practice reveals that termination is an exceptional sanction, while monetary fines are the norm. Although the Law enables the courts to issue high monetary fines (depending on the severity of the concrete criminal offense), only one monetary fine issued to date amounted to several million kunas, while the others were significantly lower.

If the presented model is observed in the context of the accelerated development and progress of technology and the intensifying presence of autonomous artificial intelligence in the economic and industrial operations, then the I conclude that the model of liability based on the liability of the responsible person will soon become inadequate. This is because it will be more difficult, if not impossible, to determine the existence and liability of a competent natural person. Therefore, I posit that future research and scholarly deliberations of the criminal liability of legal entities will have to be aimed at the development of a model of autonomous liability. For the moment, such a model is purely theoretical in nature.

**Funding:** This research received no external funding.

**Institutional Review Board Statement:** Not applicable.

**Informed Consent Statement:** Not applicable.

**Data Availability Statement:** https://podaci.dzs.hr/2021/hr/10009; https://sudskapraksa.csp.vsrh.hr/home.

**Conflicts of Interest:** The author declares no conflict of interest.

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
