# Peer review of "Corporate Criminal Liability: An Overview of the Croatian Model after 20 Years of Practice†"

_laws, 2023_

Round 1

Reviewer 1 Report

The second table 4 should be table 5

Author Response

Thank you for the Review Report. I have corrected table 4 into table 5. 

Reviewer 2 Report

The Authors, starting with a short historical outline about the reasons for the introduction of criminal liability of legal persons in the world and in Croatia, analyse the concept of legal liability of legal persons in Croatian law, and then present the issue of applying this law in judicial practice.

The analysis of the construction of criminal liability of a legal person under Croatian law is comprehensible, substantial and alludes to three models of this liability developed in both Anglo-American and European doctrine. The Authors concisely present the principles of liability of legal person in Croatian law. My only remark is that when raising the issue of the so-called effective remorse (Article 12a of the Act) they should argue: 1) whether the exemption from liability can be full or partial, 2) what circumstances justify the decision on exemption from liability (i.e. whether the efficacy of a report leading to criminal charges is reviewed etc.).

The second part of the study, concerning the practical application of the Croatian Act on legal persons liability is also valuable. The Authors analyse both: statistical data from the State Agency for Statistics of the Republic of Croatia and the State Attorney's Office of the Republic of Croatia, and the results of research on 31 final court decisions.  It is beneficial for comparative research to argue the structure of crimes committed by legal entities. Although for a reader who is not an expert in Croatian criminal law, it may be burdensome to classify certain types of crimes as "Against the Economy" (whether this is a reference to the title of a chapter in the Croatian Criminal Code, or whether it is related to the division established in Croatian doctrine, or to another distinction of this type of crime). Not every legislation distinguishes this type of crime.

However, I am not persuaded that the comments contained in point 3.3 could be precisely the issue of the application of the Act on Legal Persons Liability in practice. The Authors do not so much refer to the application of the law already in force, but indicate crucial future directions of scientific research that may develop proper legislative changes and models of application of the law in respect to the increasing use of AI technology by enterprises. Therefore, I would suggest separating this part of the study from the part concerning the practice of applying the law (part 3 of the article).

Considering the above appraisals and reasons, I recommend the publication of the reviewed study, suggesting, however, to scrutinize minor amendments indicated in the review. Although the study deals with Croatian law, it can be profitable in scientific research in other countries. The reasons indicated by the Authors for the introduction of the Act on Legal Persons Criminal Liability are typical for the countries de facto occupied by the Soviet Union after 1945, which underwent political transformation in the 1990s. Also common for such countries are difficulties with enforcing the provisions of the Act in the first period of its validity, as indicated in the study. A very interesting circumstance that gave rise to the further development of criminal liability of legal persons are the trials against Dr. Ivo Sanader – former Prime Minister of Croatia. Therefore, the Authors' remarks may be a contribution to reflections on the direction and scope of legislative changes not only in Croatia, but also in other countries where either instruments of criminal liability of legal persons have not yet developed, or laws on such liability are applied quite exceptionally.

Author Response

Thank you very much for your comments.

Effective remorse in Croatian Law does not lead to exemption from liability. Instead, it only opens possibility for the court to either mitigate or release from sentence (but this does not effect conviction for the crime). I will make that more clear.

The official title of the chapter is "Crimes against the Economy".

I will separate the last part of the study as you suggested.  

Reviewer 3 Report

The aims and scope of the work are clearly set out and the author meets the aims well. The article makes an original contribution to the literature in this area, the methodology is sound, and the analysis of the development and application of the Croatian model of corporate criminal liability provides a useful example for legal systems that have not yet developed a corporate criminal regime. Of note, this work also draws attention to the phenomenology of criminal offences committed by legal entities, ie they are predominantly economic in nature, and the increasingly urgent issue of the future use of autonomous artificial intelligence by legal entities. The need to address the harms caused by AI will become increasingly relevant for all legal systems, including those that have long established models of corporate criminality, since they typically establish corporate liability on the basis that an underlying offence has been committed by a natural person. 

The topic is covered in sufficient depth for its purpose and the references are appropriate, the fact that some (but not all) of the publications cited are over 5 years old is consistent with the area of law under consideration. The article is clearly articulated, well-structured and highly relevant for the reason given above. The conclusions drawn are consistent with the evidence provided and arguments made.

Specific comments: At lines 90-91, regarding the Anglo-American legal systems, the author may want to clarify that the objective model is characteristic of the US legal system with its principle of respondeat superior. In contrast, the English common law approach employs the identification principle whereby the liability of the legal entity is based on the liability of the of the responsible persons at the 'directing mind and will' level. (See eg the Law Commission of England & Wales, Corporate Criminality, Discussion Paper 2021 for a good overview). 

A minor point - at line 227, the author states that the number of reports over the 5 year period is even but, comparing 2021 to previous years, it is actually up by 50% (albeit based on the value of the fines imposed, the gravity of the offending behaviour is relatively low?).

Author Response

Thank you for the remarks. I will add clarifications regarding British and US systems and amend line 227.